# Notch Signaling in Skeletal Development, Homeostasis and Pathogenesis

**DOI:** 10.3390/biom10020332

**Published:** 2020-02-19

**Authors:** Jennifer T. Zieba, Yi-Ting Chen, Brendan H. Lee, Yangjin Bae

**Affiliations:** Department of Molecular and Human Genetics, Baylor College of Medicine, Houston, TX 77030, USA; Zieba.jennifer@gmail.com (J.T.Z.); Yi-Ting.Chen@bcm.edu (Y.-T.C.); blee@bcm.edu (B.H.L.)

**Keywords:** skeletogenesis, bone, cartilage, fracture repair, osteoporosis, osteoarthritis, osteosarcoma, bone metastasis

## Abstract

Skeletal development is a complex process which requires the tight regulation of gene activation and suppression in response to local signaling pathways. Among these pathways, Notch signaling is implicated in governing cell fate determination, proliferation, differentiation and apoptosis of skeletal cells-osteoblasts, osteoclasts, osteocytes and chondrocytes. Moreover, human genetic mutations in Notch components emphasize the critical roles of Notch signaling in skeletal development and homeostasis. In this review, we focus on the physiological roles of Notch signaling in skeletogenesis, postnatal bone and cartilage homeostasis and fracture repair. We also discuss the pathological gain- and loss-of-function of Notch signaling in bone and cartilage, resulting in osteosarcoma and age-related degenerative diseases, such as osteoporosis and osteoarthritis. Understanding the physiological and pathological function of Notch signaling in skeletal tissues using animal models and human genetics will provide new insights into disease pathogenesis and offer novel approaches for the treatment of bone/cartilage diseases.

## 1. Introduction

Notch signaling is an evolutionarily well-conserved pathway that regulates critical processes during embryonic development, cell fate determination, proliferation, differentiation, and homeostasis. Since the discovery of mutant flies with notched wings by Dexter and Morgan, there have been numerous studies and animal models that have elucidated the physiological and pathological roles of Notch signaling. Moreover, essential roles of Notch signaling in development have been uncovered through various inherited and somatic conditions caused by mutations in the core and accessory components of the Notch pathways. Notch receptors and their ligands are transmembrane proteins and initiate their signaling cascade by cell-to-cell physical interactions. Humans have four Notch receptors (*NOTCH1-4*) and five different Notch ligands (*JAG1, JAG2, DLL1, DLL3* and *DLL4*). Upon the interaction of Notch receptors and cognate Notch ligands via direct cell contact, Notch receptors undergo sequential proteolytic cleavages. The extracellular domain of the receptor is cleaved by metalloproteinase tumor necrosis factor-a converting enzyme (TACE) and then further cleaved by a γ-secretase complex of Presenilin1 and Presenilin2 to become a Notch intracellular domain (NICD). The NICD subsequently translocates to the nucleus and interacts with RBPJK and MAML to activate transcription by displacing the co-repressor complex bound to RBPJK. This transcription complex induces the canonical expression of Hairy Enhancer of Split family genes; *HES1, HES3, HES5* and *HES7* and HES-related with YRPW motif family genes; *HEY1, HEY2* and *HEYL*. NICD can also interact with proteins other than RBPJK to exert its function, which is termed as the non-canonical, or RBPJK-independent, Notch pathway.

The role of Notch signaling in skeletal development and homeostasis has been established by studies conducted in both genetically modified animal models and in vitro. Furthermore, a select group of genetic skeletal disorders associated with Notch mutations corroborate the physiological roles of Notch signaling in skeletal tissues.

In this review, we will focus on Notch signaling in bone and cartilage development and maintenance of homeostasis, as well as in the pathological context, such as the joint degenerative disease osteoarthritis (OA) and cancers including primary bone cancer and metastatic cancer.

## 2. Notch Signaling in Human Skeletal Diseases

The critical roles of Notch signaling were highlighted by the discovery of mutations in human genetic diseases that caused defects in development and postnatal homeostasis. Subsequently, delineating the underlying mechanisms of these diseases have illuminated the role of Notch pathways in the skeleton.

Adams Oliver syndrome (AOS), a rare congenital disorder characterized by aplasia cutis congenita and terminal transverse limb defects, is associated with mutations in several Notch pathway components. These genes include *NOTCH1, DLL4*, and *CSL*, the human ortholog of *RBPJK*. Most of the mutations occurring in *NOTCH1* reside in the EGF-like repeats of the extracellular domain, causing structural changes that lead to presumably context-dependent *NOTCH1* loss-of-function [1]. Heterozygous missense mutations in *CSL* lead to defects in DNA binding to Notch target genes [2]. Missense and nonsense mutations in *DLL4* lead to loss of *DLL4* function in AOS [3].

Alagille syndrome is an autosomal dominant disease affecting multisystem organs including the craniofacial skeleton and vertebrae. The failure of the vertebrae to fuse ventrally during development causes a ‘butterfly’ appearance in radiographic images [4]. Craniofacial developmental abnormalities include craniosynostosis and characteristic facial features, including a broad nasal bridge, deep set eyes, pointed chin, prominent forehead, and triangular facies. Also, patients have short stature, low bone mass and increased fracture incidence. Alagille syndrome is associated with loss-of-function mutations in *JAG1*, which are found across the extracellular and intracellular domain of the JAG1 protein [5,6,7,8]. Also, mutations of *NOTCH2*, either in isolation or in conjunction with mutations of *JAG1*, have been reported [9,10]. However, null mutations of *Jag1* or inactivation of *Notch2* in mice result in embryonic lethality, while *Jag1* null mice have defective embryonic and yolk sac vasculature and embryonic lethality due to hemorrhage [11]. Interestingly, the combined heterozygous inactivation of *Jag1* and a hypomorphic *Notch2* allele in mice recapitulated Alagille syndrome [12].

Spondylocostal dysostosis (SCDO) is characterized by vertebral segmentation defects and rib anomalies due to dominant, recessive, and sporadic mutations of genes encoding various components of the Notch signaling pathway [13]. *DLL3, MESP2, HES7* and *LFNG* are all found to harbor causative mutations associated with SCDO [14,15,16,17]. Inactivation of *Lfng* or *Hes7* in mice leads to abnormal vertebral columns and rib cage development [18,19]. Homozygous missense loss-of-function mutations in *LFNG* and *HES7* are associated with SCDO [17,20].

Hajdu–Cheney syndrome (HCS) is a rare, autosomal, dominantly inherited disease. Patients exhibit prominent skeletal features including facial dysmorphisms and craniofacial defects, such as micrognathia, midface flattening, and dental abnormalities. HSC is caused by heterozygous mutations in *NOTCH2* [21,22]. These mutations lead to truncated NOTCH2 protein which becomes stable to ubiquitination degradation. As a consequence, HCS mutations enhance NOTCH2 signaling. Ernesto et al. generated a *Notch2* mutant mouse harboring a truncating mutation in exon 34 upstream of the PEST domain [23]. This mouse model exhibited cancellous and cortical bone osteopenia secondary to increased bone resorption, recapitulating HSC.

Lateral meningocele syndrome (LMS) is a rare disorder characterized by facial anomalies, hypotonia, and meningocele with related neurological dysfunction [24]. Skeletal manifestations include short stature, scoliosis, pectus excavatum, Wormian bones, thick calvariae, increased density of the base of the skull, and increased bone remodeling [25]. The gain-of-function mutation in *NOTCH3* due to the deletion of the PEST domain results in elevated NOTCH3 signaling. The inheritance pattern of LMS is not yet clear. However, most documented cases reveal de novo heterozygous truncating mutations in exon 33 of *NOTCH3* [26].

In summary, the Notch pathway regulates a vast array of essential developmental processes in skeletogenesis. Not surprisingly, mutations in genes encoding the Notch pathway exhibit severe deformities and disorders as described above in Table 1. Furthermore, the fine tuning of Notch signaling can be regulated by post-translational modifications such as glycosylation and fucosylation [27]. The discovery of these genetic modifiers may help to understand the severity of these disorders. The variable expressivity in Alagille syndrome may be partially explained by these modifiers. Therefore, ongoing efforts to find human mutations by whole genome sequencing and analysis, as well as the global sharing of genomic and phenotypic data, may improve the understanding Notch-associated genetic diseases by identifying unique structure-function correlations informed by genotype as well as potentially oligogenic inheritance that is informed by strong modifier alleles.

## 3. Notch Signaling in Chondrogenesis

Appendicular skeletal development begins with a cartilage intermediate formed from the condensation of mesenchymal chondrocyte progenitors. This condensation is surrounded by a thin layer of spindle-shaped cells that will eventually form the perichondrium. Following initial condensation and proliferation, chondro-progenitors differentiate and become resting chondrocytes, after which cells undergo proliferation and develop a hypertrophic phenotype, inducing mineralization in the ECM, and finally undergoing programmed cell death. For long bones, these phases of chondrocyte proliferation and differentiation comprise a growth plate in an overall process known as endochondral bone formation. Endochondral bone formation is accomplished through the tight regulation of gene expression and multiple signaling pathways, such as TGFβ, WNT, FGF, IHH, and NOTCH. The temporal-spatial regulation of these signaling pathways determines the proper chondrocyte maturation [28,29,30,31].

The endogenous expression of Notch signaling components within the early mouse limb bud and growth plate have been well characterized. *Notch1* is expressed in regions of condensed mesenchymal cells in E12.5 embryos [32]. More specifically, Dong et al. demonstrated the spatial-temporal expression of *Notch1* throughout the limb bud mesenchyme at E11.5-12.0 and observed high levels of *Notch1* surrounding the vascular tissue [33]. *Notch2* was expressed ubiquitously throughout the limb bud. Expression of downstream targets of Notch signaling such as *Hes1* and *Hey1* were found closely localized with *Notch2* at the same embryonic stage. The expression profile of ligands *Jag1, Dll1,* and *Dll4* varied. *Jag1* was moderately expressed throughout the limb bud mesenchyme but was mainly in the distal medial mesenchyme. *Dll1* was expressed throughout the limb bud mesenchyme, and *Dll4* expression was restricted to vascular structures.

In differentiating ATDC5 cells (a murine chondrocyte cell line), Watanabe et al. demonstrated early *Notch1* expression with *Dll1* expression followed shortly thereafter [32]. As chondrocytes matured, *Notch1* and *Dll1* expression decreased [34]. The expression of Notch receptors and targets become attenuated and mainly restricted at the periphery of the cartilage condensations as chondrocytes mature [32,33]. A reduction in *Notch1* expression in human mesenchymal stem cells (MSCs) was also observed as chondrogenic differentiation progressed [35]. Subsequently, *Notch1* expression increased in the prehypertrophic and hypertrophic zones of E16.5 through E18.5 limb buds [32,36]. During postnatal long bone maturation, Crowe et al. demonstrated *Notch2* expression was observed throughout the growth plate whereas *Delta-1* expression was localized to hypertrophic chondrocytes of P8 through P10 mice [37]. Ligands Serrate1, Serrate2, and Notch1 were not detected in growth plate chondrocytes at these time points. These collective studies demonstrated dynamic expression patterns of Notch signaling components and activation, which support diverse functions throughout chondrocyte maturation and long bone formation.

Studies involving the overexpression of Notch1-NICD (NICD1) demonstrated that while Notch signaling is important for mesenchymal cell proliferation, it also represses chondrogenic differentiation. Watanabe et al. demonstrated that overexpression of NICD1 in ATDC5 cells suppressed chondrogenic differentiation by repressing expression of *Col2a1*, *Agn* as well as *Sox9*. *Hes1* overexpression in ATDC5 cells suppressed chondrocyte differentiation markers, although less effectively, indicating that additional Notch targets are involved. Furthermore, exposure of γ-secretase inhibitor to ex vivo limb bud cultures accelerated initiation of chondrogenic condensation [34], suggesting the negative role of Notch signaling during chondrogenic differentiation. *NICD1* overexpression in committed chondrocytes using *Col2a1-Cre* mice resulted in decreased proliferation of chondrocyte precursors and inhibition of sclerotome differentiation [36]. Other study demonstrated that Hey1 and Hes1 blocks Sox9 binding to the promoter region of *Col2a1*, thereby repressing *Col2a1* expression during chondrogenesis in human MSCs [35]. These studies demonstrate that Notch signaling plays a dual role in maintaining the proliferation of progenitor cells while also suppressing chondrogenic differentiation.

Subsequent studies revealed the repressive role of Notch downstream targets in chondrocyte differentiation. Dong et al. showed that Hes1 suppresses chondrogenesis in ex vivo limb bud cultures while Hey1 and HeyL are dispensable [33]. Rutkowski et al. further reported that both in vitro and *in vivo*, *Hes1* expression follows but does not affect *Sox9* expression changes during chondrogenesis [38]. However, *Hes5* had an inverse relationship with *Sox9* expression and was recruited to upstream promoter regions of *Sox9*, suggesting that *Hes5* is a negative regulator for *Sox9* transcription. Consistent with these studies, chondrogenesis was accelerated in mouse limb buds when *Hes1* and *Hes5* were deleted in limb mesenchymal cells using *Prx1-Cre* mice [38,39]. Tian et al. added to the growing list of Notch targets that affect chondrogenesis by showing NICD binds to the *Twist1* promoter region [40]. The inhibition effect of Notch on chondrogenesis was reduced with knockdown of *Twist1* in limb bud cultures.

Notch signaling also plays critical roles during chondrocyte maturation. Crowe et al. demonstrated the function of Notch in chondrocyte hypertrophy in 1999 [37]. In chick limb buds, *Delta1* mis-expression blocks chondrocyte maturation and reduces levels of ossification. The removal of γ-secretase (*Presenilin1* and *Presenilin2*), as well as the deletion of *Notch1* and *Notch2* in *Prx1-Cre* mice, resulted in the accumulation of hypertrophic chondrocytes in the growth plate during embryonic development [41]. Also, the single conditional deletion of *Notch2* but not *Notch1* recapitulated the γ-secretase removal, indicating a more predominant role for *Notch2* in growth plate chondrocyte hypertrophy [41,42]. Interestingly, Mead et al. also demonstrated an increased hypertrophic zone with the overexpression of *NICD1* in committed chondrocytes using *Col2a1-Cre* mice along with reduced *Sox9* expression [36]. *Sox9* was expressed in the proliferating zone but not in the hypertrophic zone of chondrocytes [43]. Mechanistically, it has been demonstrated that *Sox9* expression during chondrocyte hypertrophy is governed by Rbpjk. When *Rbpjk* is deleted in *Prx1-Cre* mice, *Sox9* expression is increased in prehypertrophic chondrocytes, indicating that Notch plays a suppressive role in *Sox9* expression. Furthermore, cartilage-specific *NICD1* gain-of-function mice displayed a chondrodysplasia phenotype with suppression of *Sox9* expression [44]. The removal of *Rbpjk* from this mutant mouse model rescued the limb defect but “butterfly” vertebrae persisted, suggesting that the removal of *Rbpjk* did not fully rescue axial skeleton deformity. Also, the Sox9 level was recovered upon the deletion of *Rbpjk*. Moreover, chromatin immunoprecipitation (ChiP) assays demonstrated the recruitment of Rbpjk/NICD complex to the upstream regulatory region of the *Sox9* gene, indicating a Rbpjk-dependent (Notch canonical) mechanism [44]. This study also suggested that Rbpjk-independent Notch signaling (Notch non-canonical) contributes to vertebrae development.

The function of Notch in chondrogenesis is further complicated by its ability to utilize both Rbpjk-dependent and Rbpjk-independent mechanisms, as described above. In Rbpjk-independent Notch signaling, NICD has been shown to interact with members of the BMP, TGFβ, Wnt, and NF-kb protein families, affecting various aspects of signaling [45,46,47,48,49,50,51,52,53]. Kohn et al. demonstrated that while canonical Notch signaling is important in chondrocyte proliferation and its activity decreases as chondrocytes mature, non-canonical Notch signaling from the perichondrium most likely influences growth plate chondrocyte proliferation, hypertrophy, columnar organization, and cartilage matrix turnover [54]. During limb development, Notch signaling from the perichondrium functions to limit chondrocyte proliferation and differentiation and to direct the growth of the developing limb [55]. The study identified non-canonical Notch signaling as a key regulator of perichondral bone formation using *Col2a1-Cre* mice driven NICD gain-of-function and *Rbpjk* deficiency. In this mouse model, it showed enhanced perichondral bone formation as well as changes in growth plate chondrocyte morphology. Furthermore, growth plate chondrocytes showed decreased *Ptc1* expression (a target of IHH signaling) whereas perichondral cells exhibited increased *Ptc1* expression suggesting that IHH signaling may be the link between Notch and perichondral/chondrocyte cell communication [54].

In chondrogenesis and chondrocyte differentiation, Notch signaling functions in a context-dependent manner. Although it promotes mesenchymal cell proliferation, Notch is generally a suppressor of chondrogenesis (Figure 1a). Notch signaling is typically highly expressed during early chondrogenesis and decreases as chondrocytes differentiate. As chondrocytes prepare to enter hypertrophy, Notch again plays a role as an enhancer of chondrocyte hypertrophy through the regulation of Sox9 expression (Figure 1b). The underlying Notch-mediated Sox9 transcriptional regulation has been uncovered by multiple studies mentioned above. However, Notch targets beyond Sox9 and the role of Rbpjk-independent Notch signaling during cartilage development remain to be determined.

## 4. Notch Signaling and Osteoarthritis

Osteoarthritis (OA), the most common degenerative joint disorder, is characterized by fibrosis, inflammation, loss of joint cartilage, and variable degrees of synovial tissue dysplasia [56]. As OA progresses, articular chondrocytes proliferate and acquire chondrocyte hypertrophy [57,58]. Many risk factors contribute to the development and progression of OA, including aging, obesity, traumatic injury, and genetics [59]. However, the etiology of OA is still poorly understood. Multiple studies using animal models have demonstrated that Notch signaling is one of the critical regulators in cartilage homeostasis and joint maintenance.

Notch signaling components are expressed in adult articular cartilage, and the majority of chondrocytes residing in articular cartilage express *Notch1* [60,61]. Expression levels of *NOTCH1, JAG1*, and downstream target *HES5* are low in differentiated chondrocytes. Interestingly, these genes were abundantly detected in OA biopsies [62]. In post-traumatic OA, the Notch pathway is highly activated in both human and mouse joint tissues [62]. Furthermore, when Notch signaling was blocked by the chemical inhibitor, DAPT, the proliferation of OA chondrocytes was significantly decreased.

In vivo mouse models of NICD1 overexpression within postnatal joint cartilage demonstrate the dual roles of Notch signaling in joint maintenance and OA progression [63]. Sustained Notch activation in joint cartilage leads to an early and progressive OA-like pathology. However, transient Notch activation resulted in enhancing the synthesis of cartilage matrix and promoted joint maintenance [63]. In this study, they showed that the pathological gain in Notch signaling led to increased *IL6* and *STAT3* as well as MAPK signaling, which contributed to joint cartilage degradation and fibrosis. In addition, several additional signaling pathways, such as the tyrosine kinase signaling, GPCR signaling, and NO signaling pathways, were markedly increased with sustained NICD1 expression in chondrocytes. Therefore, a fine temporal-spatial balance of Notch signaling activity is required to maintain articular cartilage homeostasis and joint integrity. Loss or reduction in cartilage-specific Notch signaling is capable of reducing the abundance of MMP13 (cartilage related protease) within murine joints, as well as delaying cartilage degeneration over the short term [64]. However, the long-term reduction in Notch signaling disrupts the normal homeostasis of the cartilage and ultimately resulted in cartilage degeneration [65]. Overall, these studies showed that Notch signaling plays a complex role in cartilage homeostasis and that transient or physiological Notch signaling in chondrocytes favors a balanced anabolic and catabolic response, whereas sustained or enhanced Notch activity elicits a pathological response through the simultaneous suppression of chondrogenic genes and the induction of genes encoding catabolic factors.

Furthermore, the deletion of *Rbpjk* in mesenchymal progenitor cells using *Prx1-Cre* showed a delay in the formation of secondary ossification centers, characterized by persistent hypertrophic cartilage in the epiphyseal region with normal articular cartilage development [65]. Interestingly, these mice later exhibited an early, progressive OA-like pathology with disruption of joint cartilage architecture and ECM composition [65]. Furthermore, to understand the contribution of postnatal chondrocytes to joint cartilage maintenance, *Rbpjk* was deleted in chondrocytes by *Col2a1-Cre*. The postnatal deletion of *Rbpjk* indeed led to age-related progressive OA development, including articular cartilage degradation, mineralization of joint cartilage and altered subchondral bone. The inhibition of Notch signaling in this study further showed altered gene expression of *Col2a1*, *Prg4* (Proteoglycan 4), *Col10a1*, *MMP13* (Metalloproteinase 13) and *ADAMTS* [65]. Overall, this study demonstrated that the *Rbpjk*-dependent Notch pathway is required for the joint maintenance and articular cartilage homeostasis. *Hes1*, a downstream target of Notch signaling, also modulates OA pathogenesis. The genetic deletion of Hes1 in postnatal articular cartilage led to delayed OA development. Mechanistically, Hes1 induces the transcriptional activation of catabolic enzymes such as *MMP13* and *ADAMTS5* by directly binding to their regulatory regions [64]. Additionally, unbiased microarray assays and ChiP-seq revealed *IL6* and *IL1rl1* as downstream targets, which are implicated in inflammation. Hes1 functions generally as a transcriptional repressor downstream of Notch signaling; however, it switches to an activator by forming a complex with CaMK2 (Ser/Thr protein kinase). This study suggested a new therapeutic potential for OA by modulating the phosphorylation of CaMK2.

Overall, these studies suggest that the role of Notch signaling in joint physiology is highly complex, as well as dose- and context-dependent. A physiological level of Notch activation is required for joint development and articular cartilage maintenance. However, the loss or sustained activation of Notch signaling leads to progressive OA development. This biphasic mode of Notch signaling in joints indeed requires fine tuning and depends on temporal-spatial context.

## 5. Notch Regulation of Osteoblast Differentiation and Osteocyte Function

Skeletal homeostasis is maintained by two distinct mechanisms: bone formation and resorption. While bone formation is driven by the differentiation of chondrocytes to cartilage and subsequent bone mineralization by osteoblasts (bone forming cells), bone resorption is a coordinated effort between osteoblast and osteoclast (bone resorbing cell) activities that remove old bone and maintain mineral metabolism. Imbalance between bone formation and resorption can cause pathological outcomes, such as osteosclerosis, osteopetrosis, or osteoporosis.

Notch signaling plays essential roles in maintaining postnatal bone homeostasis and osteoblast cell fate decisions. Initial in vitro studies of Notch in osteogenesis reflected its conflicting role in osteoblast differentiation. In genetic mouse models of Notch gain-of-function and loss-of-function at specific stages of the osteoblast lineage, Notch signaling functions via a cell-context /stage dependent manner. The overexpression of *Notch1* in osteoblast precursors, driven by the *Col1a1-3.6kb* promoter, repressed osteoblast differentiation and induced severe osteopenia [66,67]. Recent studies further demonstrated the contribution of Notch ligand *Jag1* to the regulation of the osteoprogenitor pool. Hence, the deletion of *Jag1* in early mesenchymal progenitor cells using *Prx1*-*Cre* expanded mature osteoblasts and increased bone mass [68]. In contrast to these findings, the overexpression of Notch ligand *Dll1* or the intracellular domain of Notch1 (NICD1) under the control of *Col1a1-2.3kb* promoter in committed osteoblasts resulted in a severe osteosclerotic phenotype due to the inhibition of osteoblast terminal differentiation, leading to accumulated immature osteoblasts [69,70]. These studies suggest that during the early stage of osteoblastogenesis, Notch signaling induces the self-renewal ability of mesenchymal/osteoblastic progenitor cells and suppresses their terminal differentiation to mature osteoblasts. Mechanistically, the Notch-mediated expansion of immature osteoblasts is, in part, due to the inhibition of Wnt/ β-catenin signaling by NICD1 and the suppression of Runx2 transactivation on *osteocalcin* (*Ocn*) [70]. The upregulation of *Osterix,* along with *Cyclin D* and *Cyclin E*, due to NICD expression in committed osteoblasts also leads to the expansion of immature osteoblasts [70]. Moreover, the Notch target *Hey1* mediates the inhibition of *Nfatc1*, a mediator of Notch function in osteoblasts, to suppress osteoblastogenesis accounting for the inhibition of osteoblast differentiation [42].

The loss of function of Notch signaling in committed osteoblasts by deletion of γ-secretase *Presenilin1 and Presenilin2* using *Col1a1 2.3kb-Cre* mice showed an age-related osteoporosis due to elevated osteoclastogenesis [70]. Interestingly, when *Presenilin1 and Presenilin2* were deleted in early progenitors by *Prx1-Cre* (PPS mice), these mice had a high bone mass phenotype at 2 months of age. A similar high bone mass phenotype was observed when *Notch1* and *Notch2* were deleted by *Prx1-Cre* (PNN mice) [41]. PPS and PNN mice exhibited significant bone loss with aging due to an increase in bone resorption and reduction in osteoblast numbers [41]. Moreover, the deletion of *Rbpjk* in mesenchymal progenitors resulted in a high bone mass seen in PNN and PSS mice, suggesting that canonical Notch signaling is dominant in regulating early osteoblast progenitors and osteoblast-osteoclast coupling [42].

The activation of Notch signaling in osteocytes, terminally differentiated osteoblasts, was performed by overexpressing *NICD1* using late osteoblast/osteocyte-specific *Dmp1*-*Cre*. These mice showed higher cancellous bone caused by increased bone formation and reduced bone resorption [71]. Mechanistically, the expression of *Osteoprotegerin (OPG)* was increased with a decrease in *Sclerostin* level, which led to the inhibition of bone resorption and enhanced Wnt signaling leading to high bone mass. This study indicates that Notch signaling contributes to the non-cell-autonomous function of osteocytes in the regulation of osteoclasts. Consistent with these findings, the selective deletion of *Rbpjk* under conditions of NICD1 activation, driven by *Dmp1* promoter, reversed the high bone mass phenotype [72]. These data further showed that Notch canonical signaling in osteocytes regulates osteoclast differentiation and causes osteosclerosis.

In summary, these studies conducted in conditional Notch gain/loss-of-function mouse models demonstrate two manners of regulation in osteogenesis—Notch activation during early osteogenesis sustains mesenchymal and osteoblastic progenitors by promoting proliferation, while repressing differentiation of mesenchymal cells/immature osteoblasts to terminally differentiated osteoblasts (Figure 2). Notch activation in committed osteoblasts leads to osteosclerosis by increased immature osteoblasts due to inhibition of terminal differentiation of osteoblast. When Notch signaling is activated in osteocytes, it shows high bone mass due to reducing bone resorption (Figure 2) [70,73,74,75,76,77].

Recent studies have shown several interesting mechanistic aspects of Notch function in modulating osteoblast differentiation. Notch signaling may lead to the suppression of glucose metabolism restricting osteoblast differentiation in mesenchymal progenitors [78]. Another study showed a functional link between Notch signaling and mechanotransduction in which Notch serves as a mediator in response to mechanical strain in mesenchymal cells [79]. Future work on elucidating other components in the Notch pathway, glucose metabolism, and mechanical force associated with osteogenesis and bone homeostasis will be important in understanding the broad effects of Notch on skeletogenesis.

## 6. Notch Signaling in Osteoclastogenesis and Bone Resorption

Osteoclasts are bone resorbing cells derived from differentiated monocytes and macrophages originating from the hematopoietic lineage. M-CSF (macrophage colony-stimulating factor) is essential for the proliferation and survival of osteoclast progenitors, whereas osteoclast differentiation is modulated by RANKL (receptor activator of nuclear factor-kB ligand) binding to RANK to stimulate osteoclast differentiation. OPG, a decoy receptor of RANK, binds to RANKL and inhibits osteoclastogenesis [80]. Therefore, the ratio of RANKL/OPG governs osteoclast differentiation. Following differentiation, the osteoclast begins to express TRAP, undergoes multinucleation, and begins bone-resorbing activity. Notch signaling has been shown to be a critical regulator of both osteoclastogenesis and bone resorption.

In vitro, Sekine et al. used immobilized Notch ligand Delta-1 to demonstrate the enhancement of osteoclastogenesis through Notch2/Delta-1 activation [81]. In this study, however, it was shown that the activation of Notch1/ Jag1 inhibited osteoclast differentiation. Nevertheless, inactivation of Notch signaling through γ-secretase inhibitor resulted in decreased osteoclastogenesis implying that Notch’s positive role in osteoclastogenesis is more dominant. On the contrary, another study using immobilized Delta-1 to activate Notch signaling in stromal cells showed an increase in both RANKL and OPG expression, as well as inhibition of M-CSF gene expression, resulting in reduced osteoclast differentiation [82]. Furthermore, knockdown of *Notch2* expression in RAW 264.7 cells (a murine pre-osteoclast cell line) resulted in the inhibition of RANKL-induced osteoclast formation, whereas knockdown of *Notch1* had no effect in this model [83]. The deletion of *Notch1, Notch2*, and *Notch3* receptors in mouse bone marrow macrophages increased osteoclastogenesis as well as osteoclast precursor proliferation [84]. Specifically, the loss of *Notch1* and *Notch3* enhanced osteoclastogenesis, whereas *Notch2* had no effect. Collectively, these studies indicate that the role of Notch signaling in osteoclast differentiation differs based on the specific ligand/receptor interaction. Ashley et al. demonstrated that that stimulation of Notch signaling in committed osteoclast precursors increased Notch multinucleation and resorptive activity, whereas stimulation prior to precursor osteoclast induction resulted in decreased osteoclastogenesis, demonstrating the distinctive roles of Notch based on differentiation stage [85]. Osteoclastogenesis can also be induced during inflammation by TNF and further enhanced by deletion of *Rbpjk* in osteoclast precursors [86]. The study also demonstrated that deletion of *Rbpjk* allows TNF to induce osteoclast formation independently of RANKL. This study showed the cell-autonomous role of Notch signaling in inflammatory osteoclastogenesis.

Osteoclastogenesis can also be induced in a non-cell autonomous manner through the osteoblast lineage production of RANKL and OPG. Notch signaling has a critical role in osteoblastogenesis, as discussed earlier in this review, and this role affects osteoclast formation and function as well. As was observed in PPS and PNN mouse models, their osteoporotic phenotype was due to elevated osteoclast function by increased RANKL and decreased OPG production by osteoblasts [41,70].

Overexpression of *Hes1* driven by a *Col1a1 2.3kb* promoter resulted in enhanced resorptive activity of cultured osteoclast precursors in an osteoblast-splenocyte co-culture assay [87]. This result was further supported by recent mouse models of Hajdu–Cheney syndrome demonstrating that sustained *Notch2* activation in osteoblasts results in increased osteoclastogenesis as well as osteoclast resorptive activity [23,88].

Overall, Notch activity in various stages of osteoclast differentiation is governed by specific ligand–receptor interactions through both cell-autonomous and non-cell-autonomous mechanisms. Additional studies concerning the role of receptor-ligand combination in cell types at multiple stages of differentiation would help to clarify Notch’s role in osteoclasts and aid in identifying additional targets to restore bone loss. Further, whether non-canonical Notch signaling plays a role in osteoclastogenesis and osteoclast function has yet to be determined.

## 7. Notch Function in Skeletal Stem Cells and Fracture Healing

The healing of bone fractures involves a complex series of events involving multiple tissues and signaling pathways. The process of healing consists of an initial immune response, the recruitment of progenitor cells from the bone marrow and periosteum followed by the formation of an enlarged soft callus stabilizing the injury. This cartilage intermediate then undergoes a process similar to endochondral ossification where it is replaced with woven bone that is eventually resorbed, replaced with lamellar bone, restoring the bone’s original shape. Many pathways involved in endochondral ossification also play critical roles in fracture healing, including Notch signaling. Generally, Notch signaling is upregulated upon bone injury, with *Jag1* and *Notch2* being the most highly expressed in the mesenchymal cell population and decreased during chondrogenesis [89]. In this study, it was shown that *Jag2, Notch4*, and *Hes1* had the greatest change in callus tissue gene expression, with the highest expression at 10 days post-fracture. So, while there are certain components of Notch signaling that are specifically expressed during early progenitor cell recruitment in fracture healing, there are others that are expressed throughout the healing process.

The role of Notch signaling in fracture healing has been elucidated by mouse tibial/femoral fracture models and can be divided into two groups: genetic models and treatment models. An inducible transgenic mouse model of *Mx1-Cre; dnMAML*, which impairs canonical Notch signaling by dominant negative MAML in skeletal stem cells, reduced cartilage formation in the callus due to prolonged inflammatory cytokine expression at 10 days post-fracture. These mice also showed an increased callus to bone proportion with no change in callus size along with an increase in osteoclast density but decreased osteoblast differentiation, which indicates an altered bone remodeling [90]. Targeted deletion of *Rbpjk* by *Prx1-Cre*, however, showed prominent intramembranous bone formation extending from the periosteum, but the fracture persisted in non-union. Interestingly, fracture healing was normal in animals harboring loss of *Rbpjk* in committed chondrocytes and mature osteoblasts [91]. In contrast, the activation of Notch signaling in αSMA-expressing periosteal cells inhibited their osteogenic differentiation and ability to induce ectopic bone formation [92]. These genetic mouse studies demonstrated that Notch signaling mainly compromises cell differentiation but does not affect cell proliferation during fracture repair. Clearly, Notch signaling plays critical roles during fracture repair based on healing stages as well as specific cell subpopulation.

Treatment models have also elucidated the roles of Notch in fracture healing. By treating a fracture model with an intermittent γ-secretase inhibitor, Wang et al. showed prominent cartilage and bone callus formation and superior strength compared to untreated fractures due to the promotion of mesenchymal cells differentiation [91]. Furthermore, this transient inhibition of Notch signaling also resulted in an increased osteoclastogenesis during fracture remodeling, which likely occurs in a non-cell autonomous fashion. Overall, transient Notch inhibition accelerated fracture repair. However, mesenchymal cell sheet cultures with Notch activation and allografted to fractured tissue showed enhanced callus formation by increasing osteogenic potential as well as decreasing cellular senescence [93]. The delivery of Jag1 has also been shown to enhance femoral defect healing through increased bone formation [94]. These studies reveal Notch’s ability to regulate progenitor cell differentiation in fracture healing but also reveal its opposing roles based on specific cell types as well as its signaling activation or inhibition source.

Notch signaling has important roles in angiogenesis in several tissues but more specifically during bone development. In long bone development, Notch signaling promotes endothelial cell proliferation as well as sprouting angiogenesis through its regulation of *VEGF* and *Noggin* expression [95,96]. Similar to endochondral ossification event, vascularization of the fracture callus is critical to mobilize osteoblast precursors and other progenitor cells to the fractured region. Pharmacological inhibition of angiogenesis has been shown to impair fracture healing through callus reduction [97]. Although there are no specific studies linking Notch and angiogenesis in fracture healing, it is likely that Notch plays a critical role in regulating the blood vessels invasion of the developing callus.

Overall, Notch appears to function as a suppressor of the initial inflammatory response. Meanwhile, in skeletal progenitor cells, Notch functions as a suppressor of intramembranous bone formation, but an inducer of progenitor cell commitment. Finally, in periosteal cells, Notch functions as an inhibitor of osteogenesis. Notch receptor/ligand combinations and their nuclear targets are unknown in fracture healing. While there have been some inroads made into using Notch signaling as a treatment for delayed healing, these studies are in their early stages. The connection between Notch signaling and angiogenesis has been firmly established in cancer and bone tissues, but how this connection affects fracture healing has not been fully investigated.

## 8. Pathological Function of Notch Signaling in Cancer

Notch signaling plays an essential role in determining the cell-fate decisions and differentiation of the osteoblastic lineage as well as maintaining skeletal homeostasis as described in this review. Therefore, the tight control of Notch activation is required to prevent the development of various malignancies. In particular, aberrant activation of Notch signaling in osteoblastic cells results in the development of a primary malignant bone cancer, osteosarcoma [98,99]. Consistent with these results, the expression of Notch signaling components (*NOTCH1, NOTCH2*, *JAG1*, and *DLL*) and downstream genes (*HEY1 and HEY2*) were highly upregulated in human osteosarcoma cell lines and primary human osteosarcoma samples [98,100]. Treatment with γ-secretase inhibitors in a xenograft mouse model effectively decreases tumor growth [98,101]. These findings corroborate the participation of Notch signaling in osteosarcoma pathogenesis. Notch signaling also provokes the initiation of osteosarcoma and facilitates distal organ metastasis. AN osteoblast-specific gain-of-function NICD1 mouse model (*Col1a1 2.3kb-Cre; Rosa26^NICD^*) developed spontaneous osteosarcoma and pulmonary metastasis [99]. Furthermore, the progression of osteosarcoma was accelerated in *Col1a1 2.3kb-Cre; Rosa26^NICD^* on a p53 null background. Overall, these studies demonstrated a dominant role for Notch signaling in both initiation and progression of osteosarcoma. However, the downstream mechanisms how Notch signaling promotes oncogenesis still needs to be uncovered.

Aberrant genomic integrity, including large structural chromosomal variations, is one of the key characteristics of osteosarcoma. Therefore, genomic instability has been proposed to be a causal factor leading to osteosarcoma. In the NICD transgenic mouse model of osteosarcoma described above, primary osteosarcoma cells revealed cytogenetic complexity including a high rate of aneuploidy as well as recurrent chromosomal abnormalities [99]. This observation indicates that aberrant activation of Notch in osteoblasts could induce the genomic instability found in human osteosarcoma. Previously, Fagagna’s group demonstrated a repressive role of Notch signaling in the DNA-damage response (DDR) through the inhibition of ATM signaling [102,103]. ATM signaling is one of the signature DNA damage responses and is recruited to DNA double-stranded breaks when DNA insult occurs. In their study, they showed that Notch1 competes with FOXO3a and KAT5 to form a complex with ATM. This Notch1-ATM complex inhibits ATM activation by blocking the assembly of FOXO3a-KAT5-ATM complex. This study suggested the negative role of Notch signaling in maintaining genomic stability. These collective observations of cytogenetic complexity in primary osteosarcoma and impairment of DDR by aberrant activation of Notch signaling suggest that Notch-induced genomic instability in bone could be one of the downstream mechanisms leading to bone malignancy and pulmonary metastasis. Future efforts are required to decipher the pathological role of Notch signaling in genomic instability and osteosarcoma progression.

Beyond its role in osteosarcoma, Notch signaling has been demonstrated to facilitate bone metastasis. Bone metastasis is one of the major complications of breast and prostate cancer. During the early colonization stage of bone metastasis, disseminated tumor cells reshape the osteogenic niche by establishing pathological cell-cell communication with nearby stromal cells. This niche in return provides proliferation/survival signals to foster outgrowth and hamper therapeutic efficacies [104]. Notch signaling mediates the cell–cell contact between the metastatic tumor cells and bone/stromal cells. In breast cancer, tumor-derived Jag1 induces IL-6 secretion from osteoblasts, which feeds back to cancer cells to promote tumor growth. Meanwhile, the activation of Notch signaling enhances osteoclast maturation and facilitates osteolytic bone metastasis. TGF-β released from the destruction of bone matrix also creates a feedback loop to further activate Jag1 expression in tumor cells (termed the “Vicious Cycle”) and enhance the metastatic potential of the cancer [105,106]. Similarly, in prostate carcinoma, the same mechanisms of osteolytic bone metastasis also enhance metastatic potential via the expression of *Notch1* and *Jag1* [107,108]. Moreover, the expression of *Notch1* is positively correlated with the expression of Epithelial-to-Mesenchymal markers, suggesting a role for Notch signaling in prostate cancer aggressiveness [109].

The osteogenic niche also protects metastatic tumor cells from chemotherapy. Zheng et al. discovered a tumor–osteoblast interaction that enhances the chemoresistance of breast cancer. While tumor-derived Jag1 facilitates bone metastasis, chemotherapy induces the expression of Jag1 in osteoblasts and further enhances tumor cell survival [109]. In this study, they showed that Jag1 antibody (15D11) targeting both tumor- and osteoblast-derived Jag1 helps sensitize skeletal metastases to chemotherapy in osteoblastic gain of function Jag1 mouse model.

Together, these studies demonstrated a role of Notch signaling in promoting cancer progression and establishing a bone microenvironment niche for breast cancer bone metastasis. An interesting direction to further explore the role of Notch signaling is its role in angiogenesis, bone niche to sustain cancer cell survival, maintenance of cancer stem cells, and escape of immune surveillance during bone metastasis. Future efforts in understanding Notch-mediated metastasis would also be a fundamental step for developing effective therapies to target Notch-driven cancers and counter osteogenic niche-mediated chemoresistance.

## 9. Concluding Remarks

The endeavor to understand the role of Notch signaling in skeletogenesis and skeletal homeostasis using temporal/spatial deletion and activation of Notch components in mouse models have provided important insights into skeletal conditions such as osteosclerosis, osteoporosis, chondrodysplasia, osteoarthritis and cancers. Importantly, Notch signaling exerts its function in a temporally and spatially dependent manner to maintain the multipotency and proliferation of skeletal progenitors while preventing terminal differentiation, as shown in various skeletal cells, including osteoblast, osteocyte, osteoclast and chondrocyte. Moreover, the dysregulation of Notch signaling has been observed in primary bone cancers and metastatic bone cancers and age-dependent skeletal diseases such as osteoporosis and osteoarthritis. Nevertheless, several human genetic diseases from Notch components clearly demonstrated the significant role of Notch signaling during development and postnatal homeostasis of bone and cartilage. Understanding the roles of signaling sending ligands will help us to develop therapy for skeletal disorders and cancer, although such therapies must likely take into account temporal–spatial restrictions of pathological Notch signaling. Moreover, delineating non-canonical Notch pathway regulators and modifiers will open up a new avenue of therapeutic approach for the treatment of Notch-dependent skeletal disorders.

## Figures and Tables

**Figure 1 biomolecules-10-00332-f001:**
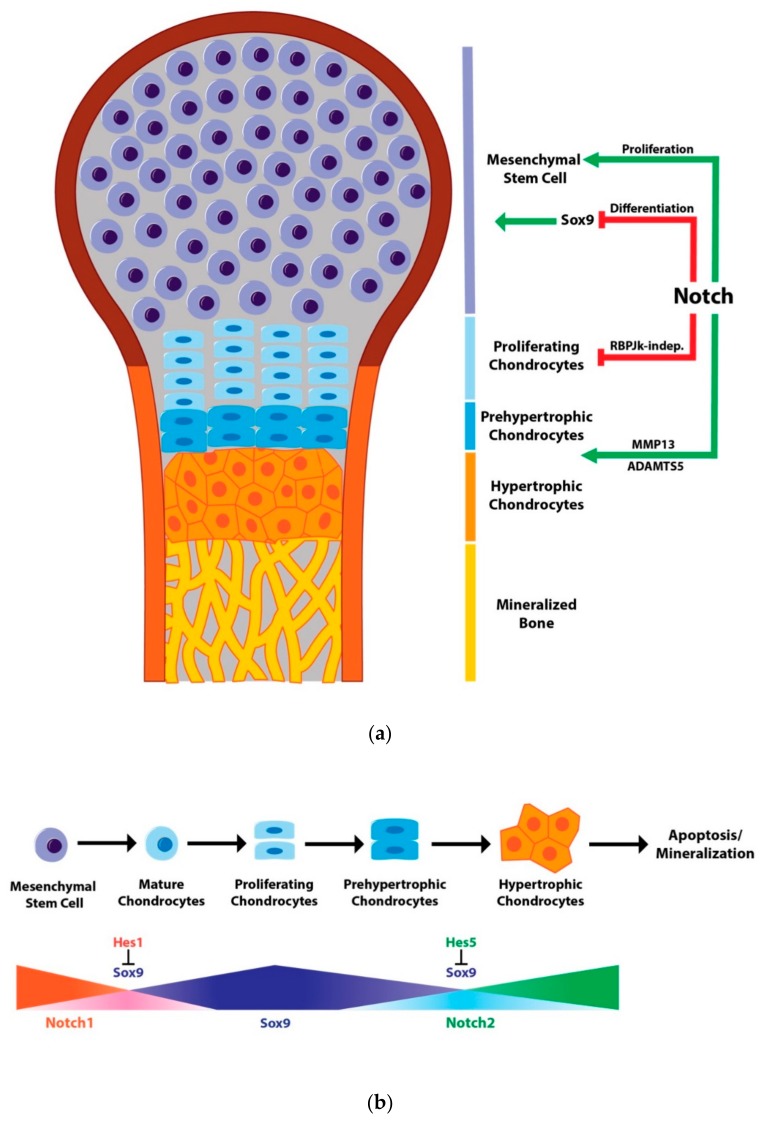
Notch signaling in chondrogenesis. (**a**) Notch signaling in growth plate development. Notch signaling maintains the mesenchymal cell population and functions to inhibit chondrogenesis through the reduction in *Sox9* as well as *Col2a1* and *Agn* transcription. As chondrocytes mature, Notch signaling promotes hypertrophy by blocking Sox9 expression while also promoting matrix catabolism through the induction of ADAMTS and MMP13 expression. Notch signaling also regulates chondrocyte proliferation and growth plate organization in a non-cell autonomous manner through its expression in the developing perichondrium. This regulation has been shown to occur independently of RBPjk; (**b**) Notch signaling in chondrocyte maturation. Notch1 expression is highest in mesenchymal progenitors and prevents chondrocyte maturation by inducing *Hes1* expression that in turn blocks *Sox9* and *Col2* transcription. Sox9 levels increase as chondrocytes mature and proliferate during growth plate maturation. This increase is then reduced by Notch2 and Hes5, allowing proliferating chondrocytes to undergo hypertrophy and eventually mineralization.

**Figure 2 biomolecules-10-00332-f002:**
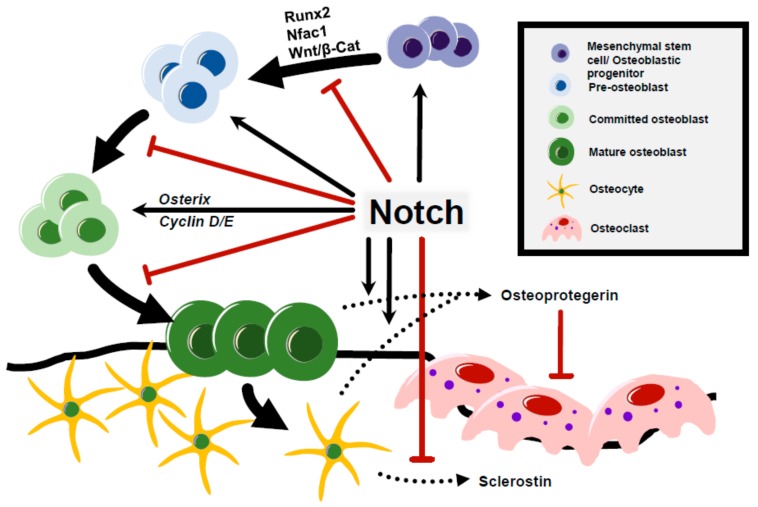
Notch regulation of osteoblast differentiation and osteocyte function. Notch maintains the mesenchymal stem cell (MSC) pool by the repression of Runx2 transactivation on *Osteocalcin* and suppresses osteoblast differentiation by the Nfatc1-mediated inhibition of Wnt/β-catenin signaling. Expression of Notch increases the immature osteoblast pool by upregulating transcription of *Osterix, Cyclin D,* and *Cyclin E*. Notch signaling regulates bone resorption in a non-cell-autonomous manner through *Osteoprotegerin* expression by osteoblasts and osteocytes. Additionally, Notch activation in osteocytes suppresses sclerostin, which enhances Wnt signaling and inhibits bone resorption.

**Table 1 biomolecules-10-00332-t001:** Human genetic diseases associated with mutations in Notch signaling.

Disease	Mutation	Notch Effect	Symptoms
Adams Oliver Syndrome	*NOTCH1* *DLL4* *RBPJK*	Loss of function	Underdeveloped skull and absent or scarred skin (Aplasia cutis congenital), mild to severe limb defect (Terminal transverse limb defects), cardiovascular malformations/dysfunctions, brain anomalies, and less frequent renal, liver and eye anomalies
Alagille Syndrome	*JAG1* *NOTCH2*	Loss of function	Multisystem disorder with a wide spectrum of clinical variability; bile duct paucity, cholestasis, cardiac defect, butterfly vertebrae, craniosynostosis and characteristic facial features, low bone mass and increased fracture incidence
Spondylocostal Dysostosis	*DLL3* *MESP2* *HES7* *LFNG*	Loss of function	Vertebral segmentation defects, rib abnormalities
Hajdu-Cheney Syndrome	*NOTCH2*	Gain of function	Short stature, coarse and dysmorphic facies, bowing of long bones, and vertebral anomalies; focal bone destruction (acroosteolysis) and osteoporosis
Lateral Meningocele Syndrome	*NOTCH3*	Gain of function	Facial anomalies, hypotonia, meningocele, short stature, scoliosis, Wormian bones, and thick calvariae

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
