# Peer review of "Notch Signaling in Skeletal Development, Homeostasis and Pathogenesis"

_biomolecules, 2020, doi:10.3390/biom10020332_

Round 1

Reviewer 1 Report

This manuscript systemically summarized the Notch function in Skeletal Development, Homeostasis and Pathogenesis.  The author also discussed the pathological gain- and loss-of-function Notch signaling in bone and cartilage, as well as in osteosarcoma and age-related degenerative diseases such as osteoporosis and osteoarthritis. This is a well written review article contains updated progress. One minor issue is that the figure 1 is not well labeled, and need to be revised with better description. 

Reviewer 2 Report

Zieba et al submit a review entitled "Notch signaling in Skeletal Development, Homeostasis and Pathogenesis". In this well written and comprehensive review they describe the role of Notch pathway and signaling in human skeletal diseases, chondrogenesis, osteoblast differentiation and osteocyte function, skeletal stem cells and fracture healing, and the pathological function of Notch signaling in cancer.

Figure 1 is not understandable in the pdf I read. It has to be completelly reworked. Please consider changing the black background; Please add a detailed legend to Figure 1. As it is a lot of the text is missing.

Please consider adding a figure with the structure of the Notch protein (various parts), and the various mutations found in human diseases clearly mapped on this structure. It would be incredibly useful for the reader.

Also a Table with the various diseases and symptoms due to these mutations.

Are these various mutations scattered on the planet, or are there are founder mutations (particular countries concerned)?

A final figure with a summarization of the role of Notch on human skeletal diseases, chondrogenesis, osteoblast differentiation and osteocyte function, skeletal stem cells and fracture healing, and the pathological function of Notch signaling in cancer

Minor.

The adress of the authors is incomplete (Town, country...)

In the first part of the manuscrit correct "et.al." to "et al"

Lots of greek symbol letters are missing; throughout the manuscript in -secetase, but also TGF , NF B... Also perhaps line 407, SMA

lines 324-326. Consider fractioning the sentence "Notch activation in committed osteoblasts and osteocytes leads to osteosclerosis by increased immature osteoblasts due to inhibition of terminal osteoblast differentiation and by promoting bone formation and reducing bone resorption (Figure 2)"

Please add a detailed legend to FIgure 2.

Round 2

Reviewer 2 Report

Changes are OK